# Desmosomal Arrhythmogenic Cardiomyopathy: The Story Telling of a Genetically Determined Heart Muscle Disease

**DOI:** 10.3390/biomedicines11072018

**Published:** 2023-07-18

**Authors:** Gaetano Thiene, Cristina Basso, Kalliopi Pilichou, Maria Bueno Marinas

**Affiliations:** Department of Cardiac, Thoracic, Vascular Sciences and Public Health, Medical School, University of Padua, 35121 Padova, Italy; cristina.basso@unipd.it (C.B.); kalliopi.pilichou@unipd.it (K.P.); maria.bueno.m@gmail.com (M.B.M.)

**Keywords:** cardiomyopathies, genetic disease, sudden death

## Abstract

The history of arrhythmogenic cardiomyopathy (AC) as a genetically determined desmosomal disease started since the original discovery by Lancisi in a four-generation family, published in 1728. Contemporary history at the University of Padua started with Dalla Volta, who haemodynamically investigated patients with “auricularization” of the right ventricle, and with Nava, who confirmed familiarity. The contemporary knowledge advances consisted of (a) AC as a heart muscle disease with peculiar electrical instability of the right ventricle; (b) the finding of pathological substrates, in keeping with a myocardial dystrophy; (c) the inclusion of AC in the cardiomyopathies classification; (d) AC as the main cause of sudden death in athletes; (e) the discovery of the culprit genes coding proteins of the intercalated disc (desmosome); (f) progression in clinical diagnosis with specific ECG abnormalities, angiocardiography, endomyocardial biopsy, 2D echocardiography, electron anatomic mapping and cardiac magnetic resonance; (g) the discovery of left ventricular AC; (h) prevention of SCD with the invention and application of the lifesaving implantable cardioverter defibrillator and external defibrillator scattered in public places and playgrounds as well as the ineligibility for competitive sport activity for AC patients; (i) genetic screening of the proband family to unmask asymptomatic carriers. Nondesmosomal ACs, with a phenotype overlapping desmosomal AC, are also treated, including genetics: Transmembrane protein 43, SCN5A, Desmin, Phospholamban, Lamin A/C, Filamin C, Cadherin 2, Tight junction protein 1.

## 1. The Discovery of Arrhythmogenic Cardiomyopathy

The discovery of arrhythmogenic cardiomyopathy (AC) dates back to the XVIII century [1] when, in 1728, the book *De motu cordis et aneurysmatibus* [2] by Giovanni Maria Lancisi (1654–1720) (Figure 1) was published posthumously.

He reported in four generations of a family exhibiting palpitations, heart failure and sudden cardiac death (SCD). The great-grandfather (first generation) died suddenly and, at autopsy, was found to have an aneurysm of the right ventricle. The grandfather (second generation) complained of palpitations of the heart, dyspnea, swollen feet and wavelike motion of the jugular veins. At dissection, the right ventricle was found to be larger than a clenched fist. The brother of the father (third generation), at over forty years old, was still living and complained of an annoying pulsation in the right side of the chest. Lancisi commented: ‘I can only hope that he may not going to end in the same way’. The proband index child (fourth generation) died of epilepsy (syncopal seizures followed by cardiac arrest?). Lancisi opened the cadaver and found the same hereditary weakness of the precordium. The four generations of the family tree were reconstructed (Figure 2) in keeping with a hereditary dominant disease. Clinical and pathological findings were pathognomic for the contemporary AC. The following assertion of Lancisi was visionary: ‘It may well be that what I had so far observed only in the right cavities of the heart can also occur in other cavities of the blood as well’.

In 1736 Lancisi’s book was republished by his heirs, with Chapter V entitled *An hereditary predisposition to cardiac aneurysm and bulging* [3].

Moreover, referring to cardiac aneurysms, Lancisi wrote: ‘An important sign of that hereditary trouble is a certain pulsation of the right side of the heart. Those who suffer from it, though in other respects are in good health, assert that they can distinctly feel this after violent physical exertion’.

In chapter VI, Lancisi mentions Hippocrates, the Father of Medicine: ‘In Hippocrates it is distinctly taught that diseases may be handed down from parents to children. Therefore no one will deny that disorders of the heart can be handed down from the very moment of conception’ [4].

Later, Gregor Mendel (1822–1884) discovered the laws of dominant and recessive inheritance through groundbreaking trait experiments.

One century later, in 1819, René Laennec (1781–1826) (Figure 3) published in Paris the book *De l’auscultation mediate*, a treatise on the diagnosis of chest diseases using the stethoscope. In Book II, Chapter XV, devoted to the accumulation of fat around the heart, he noted: ‘In medical writings we find many examples of the heart being overloaded with fat […] and even the sudden death […]. The fatter the heart is, the thinner are its walls. Sometimes these are extremely thin, being reduced almost to nothing, especially at the apex of the heart and the posterior side of the right ventricle. On examining ventricles, the scalpel seems to reach the cavity without encountering almost any muscular substance’ [5].

The description of Laennec is unequivocal: the lesions are located in topographic sites, known nowadays as the ‘triangle of dysplasia’ [7] and confirmed by subsequent pathological studies [8].

George Eliot (1819–1880), in 1871, published the book *Middlemarch*, where the protagonist Dr. Lydgate, talking with one of his patients, said: ‘I believe that you are suffering from what is called fatty degeneration of the heart, a disease which was first divined and explored by Laennec, the man who gave us the stethoscope, not so many years ago. It is my duty to tell you that death from this disease is often sudden. At the same time, no such result can be predicted’ [9].

The concept of fatty heart (=adipositas cordis) was also known to Charles Dickens (1812–1870), who described an obese, dyspnoeic and sleepy boy (‘the fatty Joe’) in his famous book *Pickwick Circle* of 1836–1837 [10].

In the classical textbook of pathology, “*Treatise of special pathological anatomy*”, published in 1896 by Eduard Kauffman (1860–1931), lipomatosis, adipositas or obesitas cordis (‘Fettherz’) of the RV received peculiar attention, being observed in subjects dying suddenly [11]. Nowadays, adipositas cordis is classified separately from AC [12].

In 1905, William Osler (1849–1919) (Figure 4A) reported in the VI edition of his famous treatise *The principles and practice of Medicine*, an amazing heart specimen that is now in the McGill College Museum, showing a “parchment-like” thinning of the ventricular walls, uniform dilatation of right auricle and right ventricle, with only epicardium remaining [13] (Figure 4B).

The pathologist Maude Abbott (1869–1940), curator of the McGill College Museum, found the aforementioned specimen among those donated by Osler during his professorship in Montreal (1874–1884). Unfortunately, the specimen was not accompanied by clinical and autopsy records. She vaguely remembered that it belonged to a man who had died suddenly while climbing a steep hill. Abbott, in the book of her memoirs written by MacDermot in 1941, told that Osler, having returned to Montreal and visiting the McGill College Museum, might have seen the specimen again and mentioned it in the new edition of his treatise [14].

Harold Segall (1897–1990) had the opportunity to examine the original specimen, preserved in a formalin-filled jar, which appeared as a large cyst (Figure 4B). The coronary arteries were patent so that an ischaemic substrate could be ruled out. Both ventricular walls appeared “parchment-like”. The ventricular septum was spared, and the heart weight was only 168 g. Histology disclosed very rare cardiomyocytes in paper-thin ventricular walls. Segall advanced the hypothesis of myocardial dystrophy in a person with generalized muscular dystrophy, such as Duchenne and Becker diseases. However, it is difficult to believe that a patient, able to climb a hill, could be affected by a generalized skeletal muscle dystrophy. He coined the term ‘Osler parchment heart’ [15].

A controversial case, which has been the source of subsequent misconceptions, was the one observed in 1952 by Henry Uhl (1921–2009) at the Johns Hopkins Hospital in Baltimore in an 8-month-old female infant who died of congestive heart failure in the absence of arrhythmias. The case was published with the title *A previously undescribed congenital malformation of the heart: almost total absence of the myocardium of the right ventricle* [16]. The description deserves to be mentioned: ‘Externally the heart appears greatly enlarged, almost the entire dilated chamber (RV) was occupied by a large laminated mural thrombosis which adhered firmly to the endocardium along the anterior wall of the ventricle. Examination of the cut edge of the ventricle wall revealed it to be paper-thin with no myocardium visible. In the RV wall, epicardium and endocardium lay adjacent to each other with no intervening cardiac muscle. No fibro-fatty tissue in the RV free wall was observed’ (Figure 5). The early age of the infant and the peculiar pathological description point to a structural heart disease present at birth (congenital anomaly), as emphasized in the title itself. Whether the disease was a genetically determined AC developed during the fetal period or not remains intriguing. The clinical phenotype was characterized neither by cardiac arrhythmias nor by a family history of heart disease.

## 2. Contemporary History of Arrhythmogenic Cardiomyopathy

Adult cases with paper-thin ventricular walls (including Osler’s case) unfortunately have been reported with the eponym of Uhl’s anomaly, clearly a misnomer because the parchment heart in adults is the end stage of a late progressive loss of the myocardium followed by fibro-fatty replacement [8,12].

French investigators (Robert Froment (cardiologist) and Robert Loire (pathologist)), reported in 1968 cases of ‘ventricule droit papyrace’ ecg [17]. They were the first to demonstrate that fibro-fatty infiltration of the RV was associated with inverted T waves in the right precordial leads of the ECG.

On the contrary, the cases reported in the literature of infants under the age of 15 months with the eponym of Uhl’s anomaly all featured congestive heart failure and isolated RV involvement (whether segmental or diffuse) without arrhythmias, in keeping with the original description by Uhl [12,18].

Since the early 1960s, the University of Padua set milestones in the history of AC. Sergio Dalla Volta, Professor of Cardiology, described ‘auricularization of the right ventricular pressure’ in six cases using right cardiac catheterization in the absence of an effective RV contraction. The blood was directly pushed into the pulmonary artery by atrial systole [19]. The autopsy finding in two cases with ‘sclerosis of the right ventricle’ was interpreted as a consequence of myocardial infarction without coronary obstruction, clearly a misdiagnosis [20]. The concept of non-ischaemic cardiomyopathy was still to be conceived. Interestingly enough, out of the six documented cases, ranging from 21 to 40 years old, two had clinically ventricular arrhythmias, five inverted T waves in the precordial leads and four congestive heart failures [20]. One of the latter, a 21-year-old woman in 1996, developed ventricular tachycardia with left bundle branch block (LBBB) morphology and right ventricular failure. She underwent cardiac transplantation, and the removed heart showed an extremely dilated RV with a parchment-like wall (Figure 6).

In 1970 Vito Terribile Wiel Marin (1939–2015), a cardiac pathologist at the Institute of Pathological Anatomy, University of Padua, performed a post-mortem of a 43-year-old lady with a clinical history of palpitations and congestive heart failure, who died of pulmonary thromboembolism. In the autopsy report, he described extreme dilatation, fibro-fatty replacement and mural thrombosis of the RV in association with left ventricular ‘myocardial sclerosis’ (Figure 7), all features in keeping with what today we call biventricular AC.

## 3. AC Is a Heart Muscle Disease with Peculiar Electrical Instability

Guy Fontaine, in the late-1970s, realized that the right ventricle (RV) might be the source of ventricular arrhythmias, with LBBB morphology on the ECG [21,22].

In 1982, Frank Marcus et al. reported a series of adult patients [7] affected by a new syndrome characterized by a remodeling of the RV, with aneurysms located in the inflow, apex and outflow, due to a fibro-fatty replacement, which they called right ventricle dysplasia, believing it to be due to a cell dysplastic congenital phenomena (Figure 8). The ECG became fundamental for diagnosis, with inverted T waves in the right precordial leads, wide QRS, post-excitation epsilon wave due to delayed electrical impulse transmission in the RV outflow tract (“late potentials”) (Figure 9), premature ventricular beats and ventricular tachycardia with LBBB morphology.

## 4. Pathological Substrates

Pathological studies demonstrated fibro-fatty replacement of the RV-free wall, starting in the subepicardium and extending along the wavefront to the subendocardium (Figure 10).

Microscopic investigation showed a loss of the myocardium, with a fibro-fatty replacement that was frequently biventricular (Figure 11), as a consequence of an ongoing, non-ischemic myocardial cell death and repair [8] (Figure 12).

Electron microscopy demonstrated disruption of the intercalated disc as a final common pathway of cell death [24] (Figure 13). Cardiomyocyte death occurs in the form of apoptosis [25] (Figure 14), associated with myocardial inflammation (Figure 15). Whether the latter is a reaction to cell death [26] or an immune phenomenon [27] remains controversial.

The origin of the adipocytes is neither a fatty-tissue infiltration from the subepicardium (adipositas cordis) [12] nor a fatty metaplasia of cardiomyocyte [28].

Mesenchymal cells are the source of adipocytes and fibroblasts, accounting for fibro-fatty tissue repair of cardiomyocyte death [29]. The phenomenon resembles dystrophy more than myocarditis or congenital dysplasia [8].

## 5. Nomenclature and Classification

Different terms have been employed in the past to give a name to this heart muscle disease: right ventricular dysplasia [7], right ventricular cardiomyopathy [30] and arrhythmogenic right ventricular cardiomyopathy/dysplasia [31].

In 1996, heart disease was definitively introduced in the classification of cardiomyopathies by the World Health Organization with the name arrhythmogenic right ventricular cardiomyopathy [32]. With the discovery of the left ventricular variant, the term arrhythmogenic cardiomyopathy (AC or ACM) has been introduced [33].

## 6. Arrhythmogenic Cardiomyopathy as a Cause of Sudden Cardiac Death

On 14 May 1979, a young cycling champion died suddenly during a tennis match in Mirano, Venice (Figure 16). He stopped playing, took his pulse, moved towards the back of the tennis court, collapsed and died suddenly. Transmural fibro-fatty replacement of the RV-free wall was found at autopsy. He was a physician, and a detailed note was found in his diary, dated 4 October 1978: ventricular tachycardia with LBBB (Figure 16). It was an ECG taken during an episode of palpitation. He represents ‘patient zero’ of a series of sudden deaths by AC in young adults published in the New England Journal of Medicine [30] (Figure 17A). The paper was the first description of a novel disease responsible for causing sudden death in the young (Figure 17B).

Sports activity was proven to increase the risk of SCD [34]. Much lower prevalence was reported from other countries [35,36,37], probably because of misdiagnosis at post-mortem. The AC rate in Italy for sudden deaths in athletes was 27% (Figure 18), which showed a sharp decline with the use of ECG screening for sports eligibility [38] (Figure 19).

## 7. Arrhythmogenic Cardiomyopathy: A Genetically Determined Heart Muscle Disease

A dominant form of AC was reported in the area of Piazzola, in the Veneto Region, by Andrea Nava, the leader of our team [39]. For years AC was known by the nickname “Venetian disease” as a local genetically determined cardiomyopathy. The clinical phenotype was proven to be absent at birth and becoming overt at 10–14 years of age [40].

A recessive form of AC, with keratoderma and woolly hair (cardiocutaneus syndrome), had been reported in 1986 from the Naxos island in Greece [41] (Figure 20), the island where the Greek mythology tells that Arianna was left by Teseo.

A race started to discover the gene of AC. In 1996, Ruiz et al., studying *junctional protein* (*JUP*) in knock-out mice, discovered that *JUP* absence influences the development of desmosome in the heart and that the human gene is located in chromosome 17q21 [42].

In 1998, Coonar et al., by linkage analysis, mapped the locus of the Naxos disease gene in humans to chromosome 17q21 [43]. Finally, in 2000, McKoy et al. identified a deletion of the *JUP* gene in patients with Naxos AC [44].

In Ecuador, the group of dermatologists Carvajal–Huerta found a recessive mutation of *desmoplakin* (*DSP*) in a family with dilated cardiomyopathy and cardiocutaneous syndrome [45,46].

The pathological study of the heart of a child of an affected family, who died of congestive heart failure, revealed a biventricular AC with a triangle of dysplasia of the RV [47] (Figure 21).

*DSP* became a candidate also for the dominant AC. The molecular genetic investigation carried out in Venetian families revealed mutations of human *DSP* [48]. Genotype–phenotype correlations showed biventricular involvement [49] (Figure 22).

A cascade of mutations in genes encoding desmosomal proteins was then discovered in dominant AC families: *plakophilin-2* [50], *desmoglein-2* [51], *desmocollin-2* [52,53] and also *plakoglobin* [54] (Figure 23), confirming that AC is a desmosomal disease. Genetic screening of the proband to unmask asymmetric carriers turned out to be life-saving.

Multiple compound or heterozygous mutations were proven to entail a more severe prognosis [55]. The disease was reproduced in transgenic mice [56,57] (Figure 24) and in zebrafish [58,59,60]. However, electrical instability was found in the absence of a pathological substrate [61].

## 8. Non-Desmosomal Arrhythmogenic Cardiomyopathy

Over the years, genetic variants other than desmosomal genes have been reported to be associated with the AC phenotype. In this setting, *Transmembrane protein 43* (*TMEM43*) was linked to the disease in 2008 with the founder mutation p.(Ser385Leu) in Canada [62], with a fully penetrant pattern. More recently, other rare genetic variants in *TMEM43* have been published [63,64]. The cardiac sodium channel gene *SCN5A* was reported in a case of an AC patient in 2008 [65] and, subsequently, in other studies [66,67]. *Desmin* (*DES*) was also associated with AC in patients with the founder mutation p.(Ser13Phe) [68,69]. Over the years, other genetic variants in *DES* have been reported [70,71,72,73,74,75] that are associated with the left ventricular phenotype. *Phospholamban* (*PLN*) has been linked with the pathogenesis of both AC and dilated cardiomyopathy (DCM) in p.(Arg14del) carriers [76]. Multiple studies demonstrated that this variant is a founder mutation widespread in the Dutch population [77,78,79,80]. In 2012, *Lamin A/C* (*LMNA*), usually associated with DCM, was also linked to AC phenotype [81], followed by other studies [82,83,84]. Truncating variants in *Filamin C* (*FLNC*) were associated with AC for the first time in 2016 [85], and since then, rare genetic variants have been found in several cases [86,87,88], specifically associated with left-ventricular phenotype and characterized by late-onset presentation with typical ECG and CMR features [89].

Several genes have been associated with AC over the years due to the evolution of genetic analysis with next-generation sequencing technology. However, genetic screening in large populations of recently associated genetic variants to AC is still missing.

In 2017, two independent studies reported for the first time rare genetic variants in *Cadherin 2* gene (*CDH2*) in AC patients [90,91]. Further, a recent multicentric study reported *CDH2* genetic variants in a large AC cohort [92]. In 2018, the *Tight Junction Protein 1* gene (*TJP1*) was linked once to the disease phenotype in a mixed cohort of 40 Italian– Dutch–German AC patients [93]. Similarly, *Integrin-linked kinase* (*ILK*) was first associated with AC in knockout mice, and two missense variants were subsequently found in two families with incomplete penetrance [94,95]. Instead, a homozygous missense variant in the *LEM domain-containing protein 2* (*LEMD2*), associated with juvenile cataracts, was also linked to a unique form of arrhythmic cardiomyopathy [96]. As described in the review of Stevens et al., [97] more than ten non-desmosomal genes have been linked to AC, including *Catenin alpha 3* (*CTNNA3*), *Titin* (*TTN*) and *Ankyrin 2* (*ANK2*). However, pathogenic variants in these genes have been reported in a minority of AC patients (1–3%) [98].

The overwhelming quantity of data derived from large-scale genetic screening is leading to a huge number of variants of unknown significance resulting in “genetic noise” without clear evidence. To this regard, a recent international reappraisal of genes associated with AC has been addressed by the National Institutes of Health (NIH)-funded resource ClinGen, showing that only eight genes show definite to moderate evidence for AC, and among them, five genes encoding for the desmosomal proteins and *TMEM43*, *DES* and *PLN* [99].

## 9. Advances in Clinical Diagnosis

Diagnostic criteria for in vivo diagnosis were first put forward in 1994 [100] and updated in 2010 [101]. ECG and echocardiography were crucial to unmask electrical instability and mechanical dysfunction. Angiocardiography was originally employed as a gold standard to detect wall dyskinesia and aneurysms in the triangle of dysplasia [102] (Figure 25).

Endomyocardial biopsy (EMB) was implemented to detect in vivo the pathognomic substrate, thanks to the transmural fibro-fatty replacement [103] (Figure 26). EMB plays a crucial role in the differential diagnosis of overlapping diseases, like dilated cardiomyopathy, myocarditis, sarcoidosis and idiopathic tachycardia of the RV outflow.

The advent of cardiac magnetic resonance (CMR) with late enhancement gadolinium facilitated the detection of not only morpho-functional abnormalities but also tissue alterations. The use of CMR unveiled isolated LV involvement in the form of subepicardial fibro-fatty ‘scars’ [104] (Figure 27). A novel mutation of desmoplakin was found in patients with LV variants of AC [105].

As far as electrophysiology, electroanatomic mapping was invented tand used to discover fibro-fatty scars in vivo with electrical silence [106] (Figure 28).

The inverted T wave in the precordial leads has been confirmed to be a pathognomic marker of the disease [107].

## 10. Prevention of SCD

Implantable cardioverter defibrillators (ICD) have proved to be effective lifesaving devices in subjects with AC to prevent SD [108] (Figure 29). Indication for implantation depends upon risk stratification [109]. It is mandatory in patients with AC and a history of cardiac arrest, unexplained syncope and sustained ventricular tachycardia (Figure 30).

The availability of automatic external defibrillators should be increased in public places, playgrounds and even at the homes of AC patients. The training of lay people to use is mandatory.

## 11. Research Globalization

The key to success in AC knowledge development was the result of an interdisciplinary approach and international collaboration. Many foreign scholars visited the University of Padua, which was considered the “mecca” of AC. Grants from the European Commission and NIH were fundamental for research and the discovery of AC genes and for setting up registries.

European meetings were held in Naxos, Baltimore, Denver and Padua, and a fruitful intercontinental collaboration was established that resulted in the publication of a monograph in 2007 [110].

Meanwhile, a great many of the main protagonists regrettably have passed away: Camerini, Dalla Volta, Fontaine, Marcus, Moss, Nava, Protonotarios and Rossi. They will remain unforgettable.

## Figures and Tables

**Figure 1 biomedicines-11-02018-f001:**
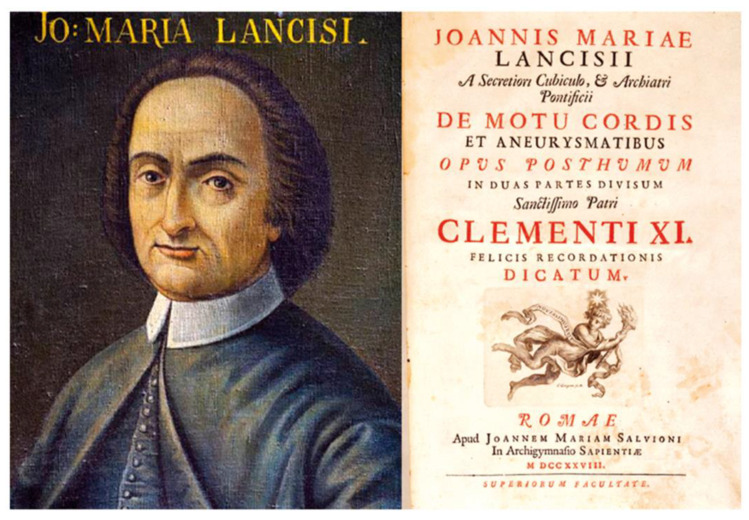
Portrait of Giovanni Maria Lancisi (1654–1720) and title page of his book *De motu cordis et aneurysmatibus*, published posthumously in 1728. From [1].

**Figure 2 biomedicines-11-02018-f002:**
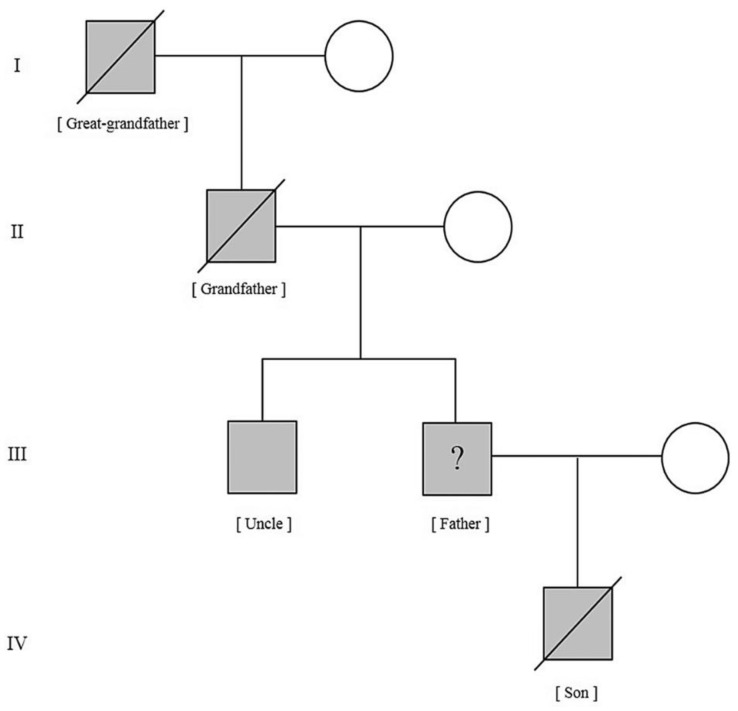
A four-generation family tree, reported by Lancisi, in keeping with a hereditary dominant genetically determined disease. From [1].

**Figure 3 biomedicines-11-02018-f003:**
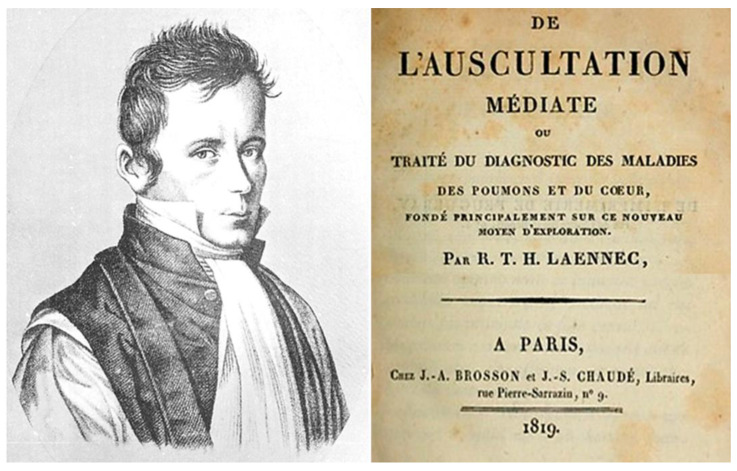
Portrait of René Laennec (1781–1826) and the title page of his book *De l’auscultation mediate*, 1819. From [6].

**Figure 4 biomedicines-11-02018-f004:**
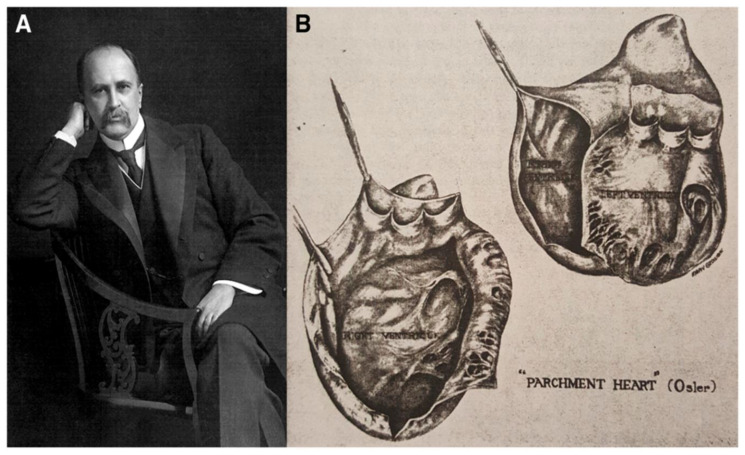
(**A**) Portrait of William Osler (1849–1919). (**B**) The Osler parchment heart. From [1].

**Figure 5 biomedicines-11-02018-f005:**
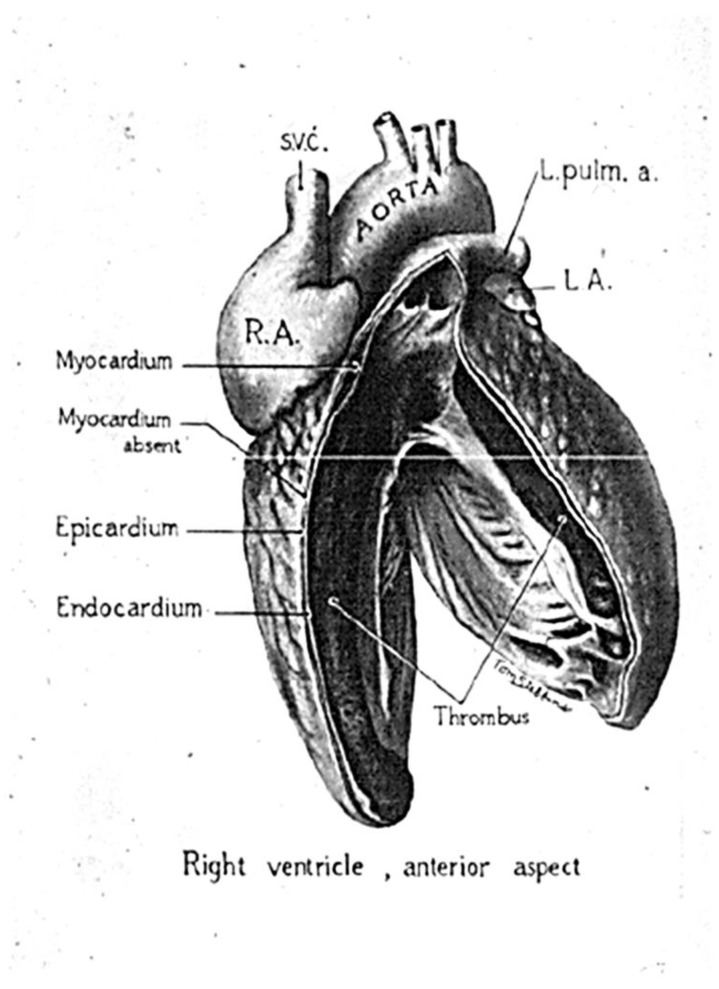
The heart reported by Henry Uhl (1921–2009) with almost total absence of the myocardium of the right ventricle. From [1].

**Figure 6 biomedicines-11-02018-f006:**
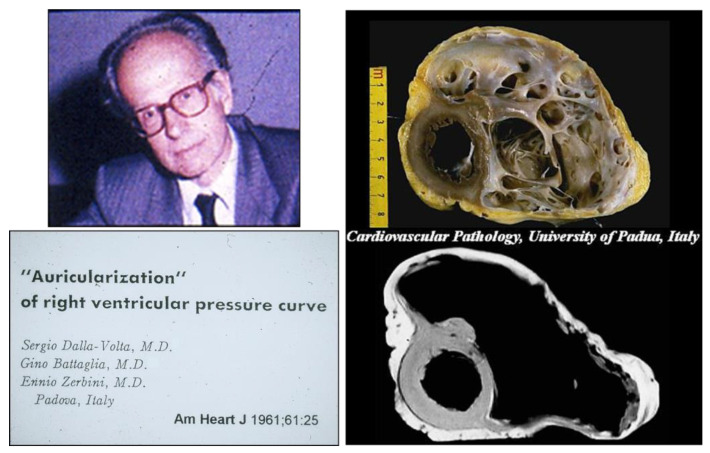
The case published by Sergio Dalla Volta in 1961, with huge dilatation of the right ventricular cavity and paper-thin free wall. In 1996 the patient underwent cardiac transplantation to treat congestive heart failure. From [1], in part.

**Figure 7 biomedicines-11-02018-f007:**
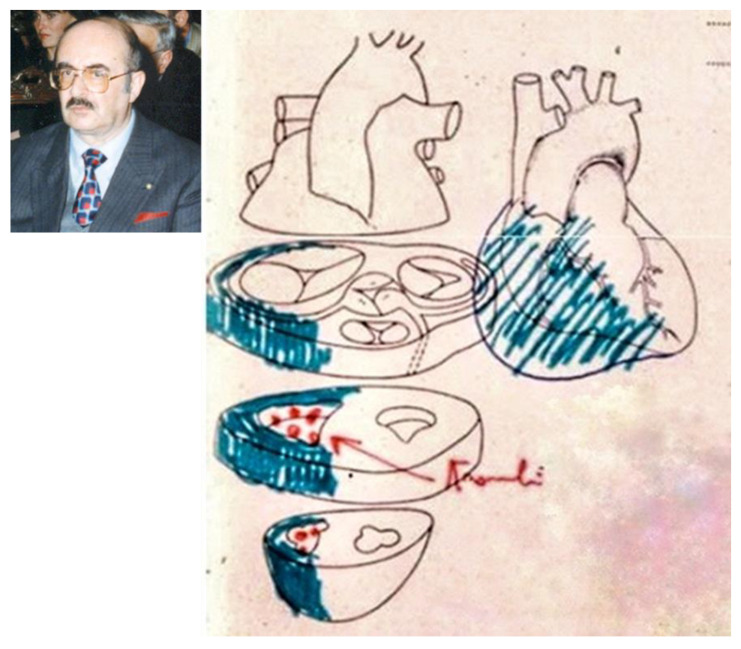
An autopsy case of arrhythmogenic cardiomyopathy of a patient that died in 1970 from pulmonary thromboembolism with mural thrombosis and fibro-fatty replacement of the right ventricle (original drawing from the autopsy report). The red arrow indicates endocardial thrombi, the blue means fibro-fatty replacement. From [6], modified.

**Figure 8 biomedicines-11-02018-f008:**
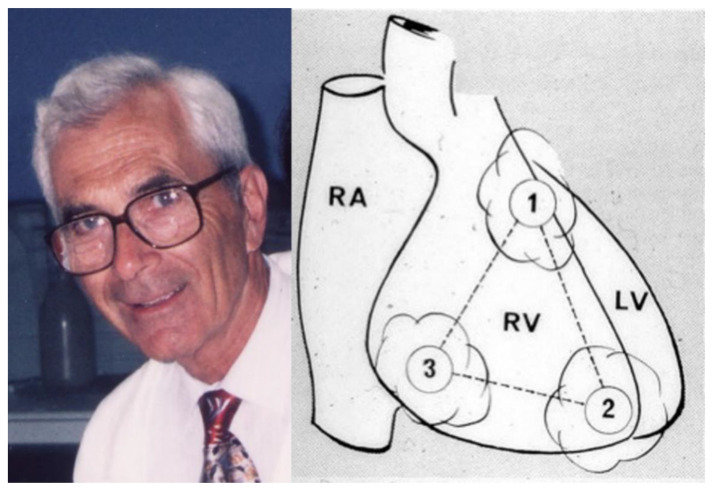
Frank Marcus and the triangle of right ventricular dysplasia with aneurysm of the right ventricle (1, 2, 3). From [23], modified.

**Figure 9 biomedicines-11-02018-f009:**
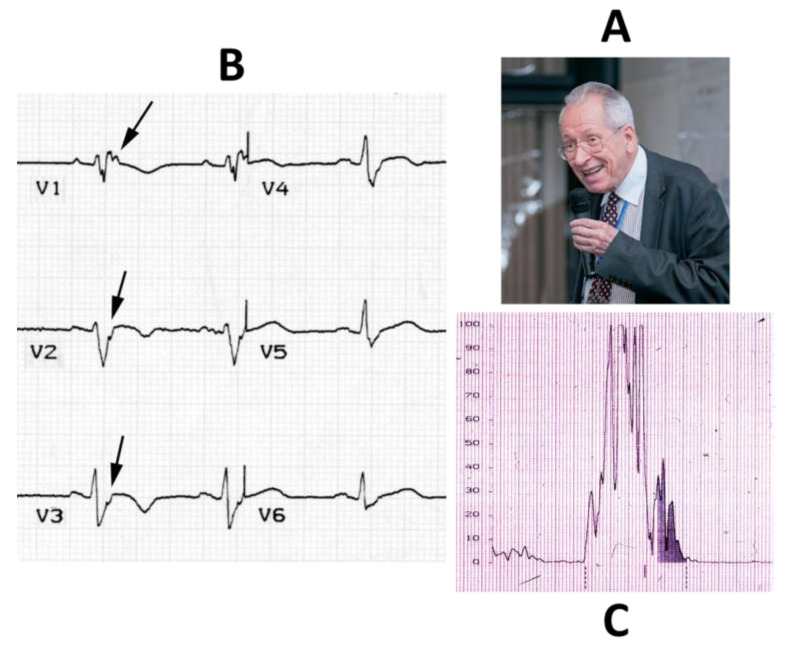
Guy Fontaine (**A**), epsilon waves (arrows) (**B**) and late potentials (**C**).

**Figure 10 biomedicines-11-02018-f010:**
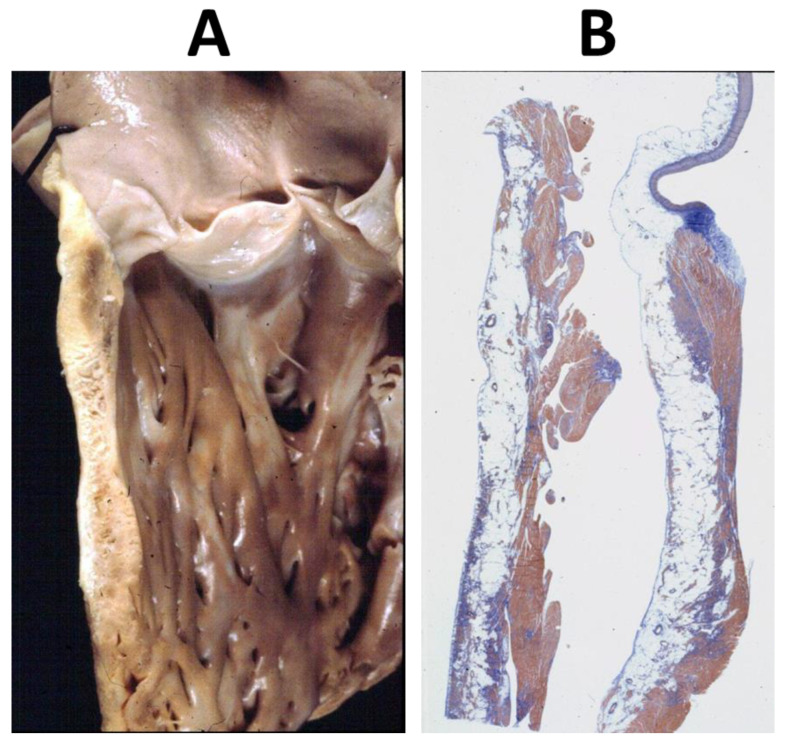
Sudden death in a 17-year-old boy: fibro-fatty replacement of the infundibular right ventricular free wall, extending along the wave-front from the epicardium to the endocardium (**A**,**B**). Azan Mallory, original magnification ×3.

**Figure 11 biomedicines-11-02018-f011:**
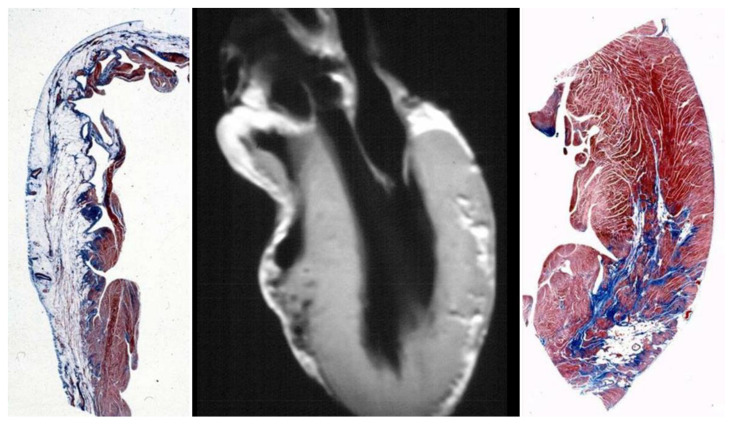
Arrhythmogenic cardiomyopathy by biventricular involvement with fibro-fatty replacement. Note that the interventricular septum is spared. Azan Mallory, original magnification ×4.0. From [6].

**Figure 12 biomedicines-11-02018-f012:**
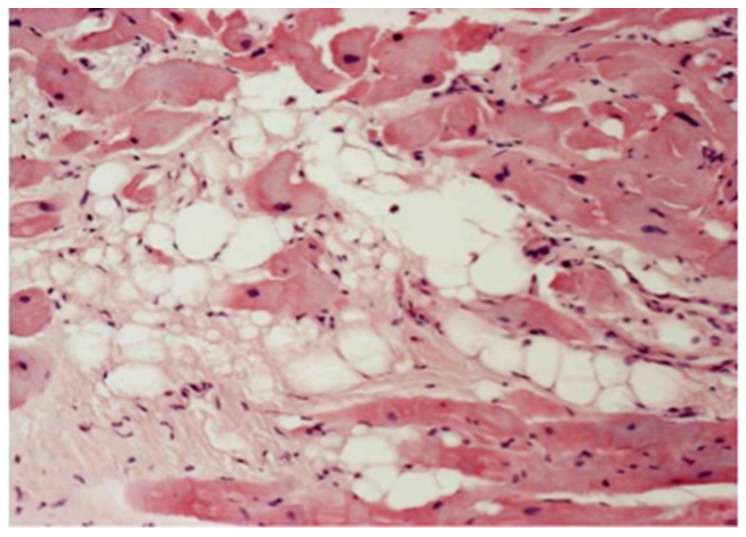
AC: close up of fibro-fatty tissue replacing dead myocardium. Hematoxylin–eosin, original magnification ×60. From [23], modified.

**Figure 13 biomedicines-11-02018-f013:**
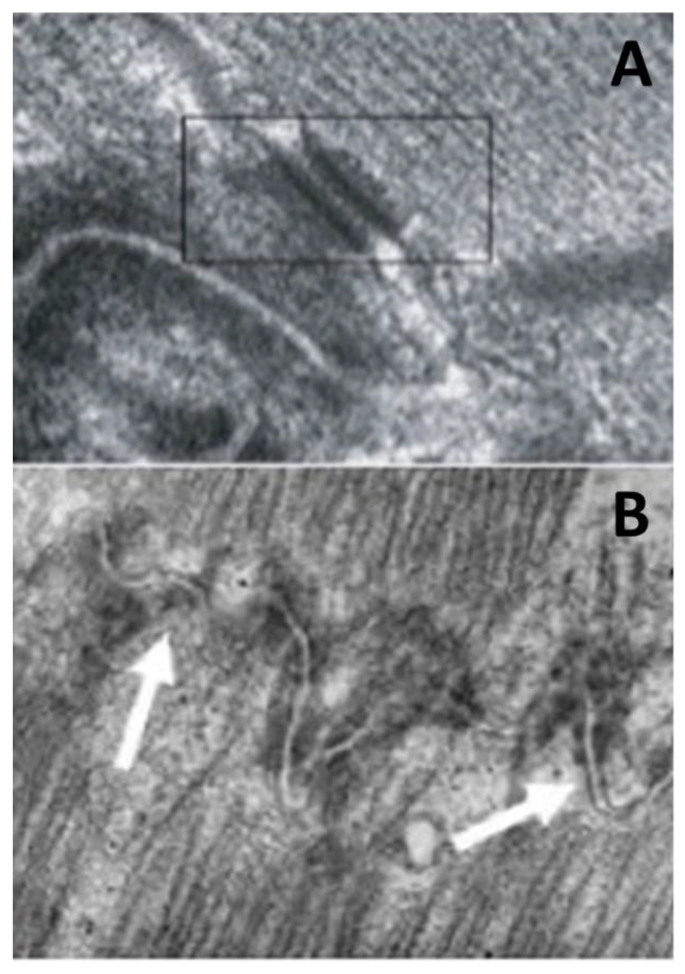
Transmission electron microscopy of disrupted intercalated disc in AC (**B**) compared to desmosome in a cavital (**A**). Original magnification, (**A**) ×30.000; (**B**) ×5. The rectangular block includes a normal desmosome, whereas arrows indicate disrupted desmosomes in AC.

**Figure 14 biomedicines-11-02018-f014:**
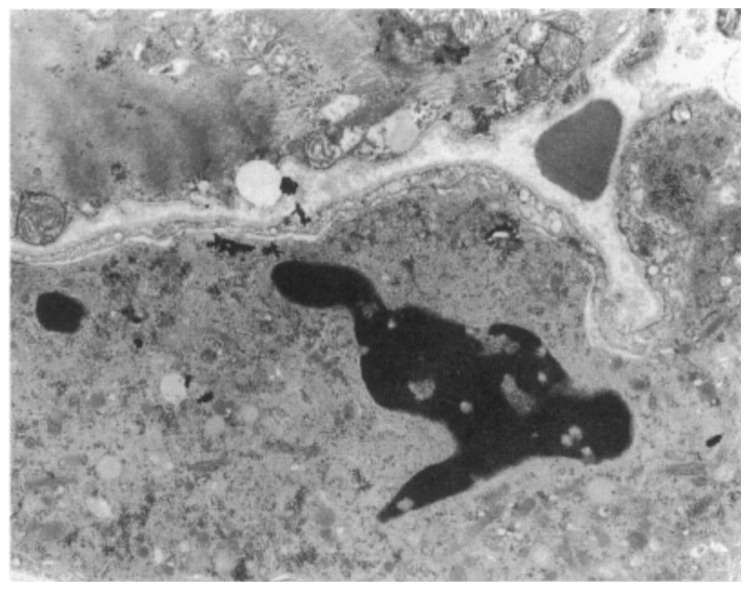
Transmission electron microscopy of myocardial apoptosis in AC. Original magnification, ×14.500. From [25].

**Figure 15 biomedicines-11-02018-f015:**
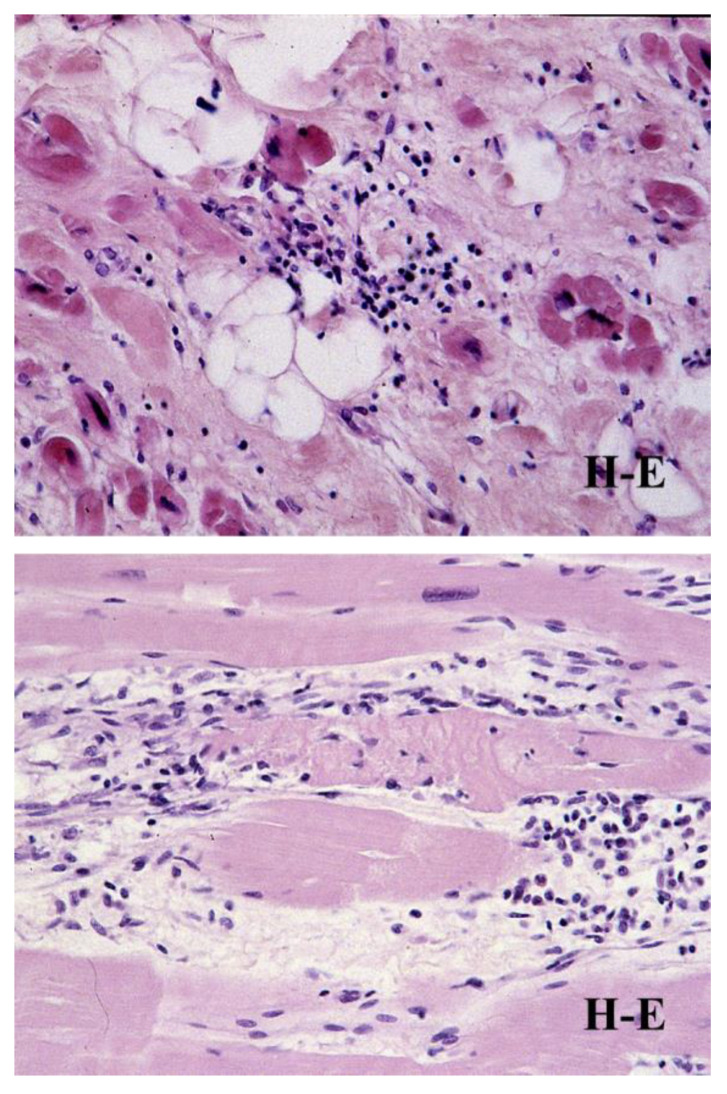
Myocarditis and cell death associated with fibro-fatty replacement in AC patient. Hematoxylin-eosin, original magnification ×40.

**Figure 16 biomedicines-11-02018-f016:**
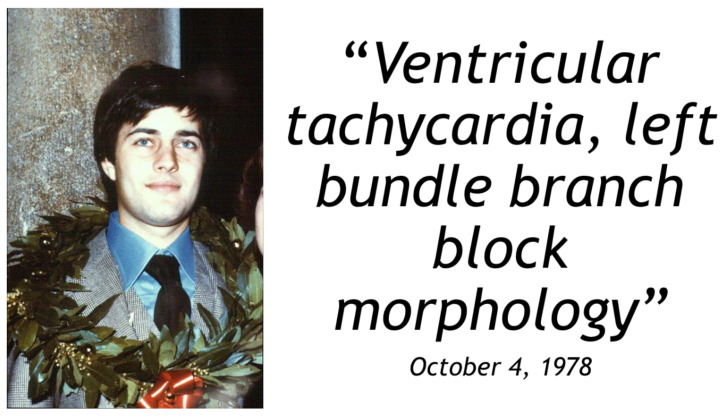
Case “0” of the Veneto Region experience of sudden death by arrhythmogenic cardiomyopathy, with the note in his diary. He was a young physician who died suddenly during a tennis match. From [23], modified.

**Figure 17 biomedicines-11-02018-f017:**
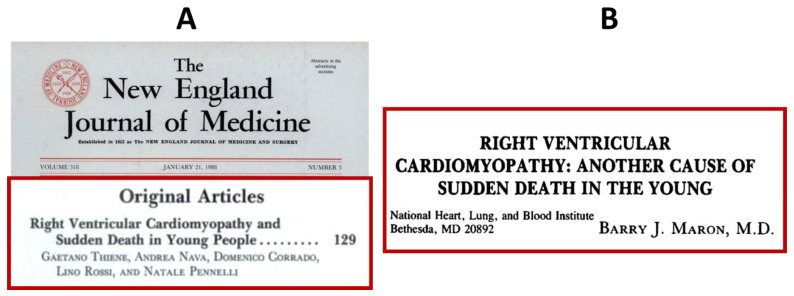
Publication in the New England Journal of Medicine 1988 (**A**) of AC, as a new disease causing sudden death in the young (**B**).

**Figure 18 biomedicines-11-02018-f018:**
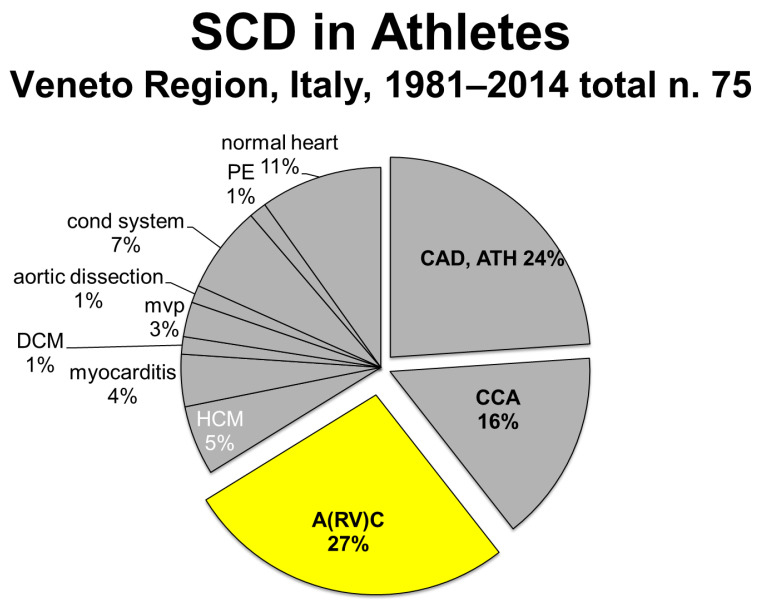
Arrhythmogenic cardiomyopathy (A(RV)C) appears to be the first cause (27%) of sudden cardiac death (SCD) in the Veneto Region.

**Figure 19 biomedicines-11-02018-f019:**
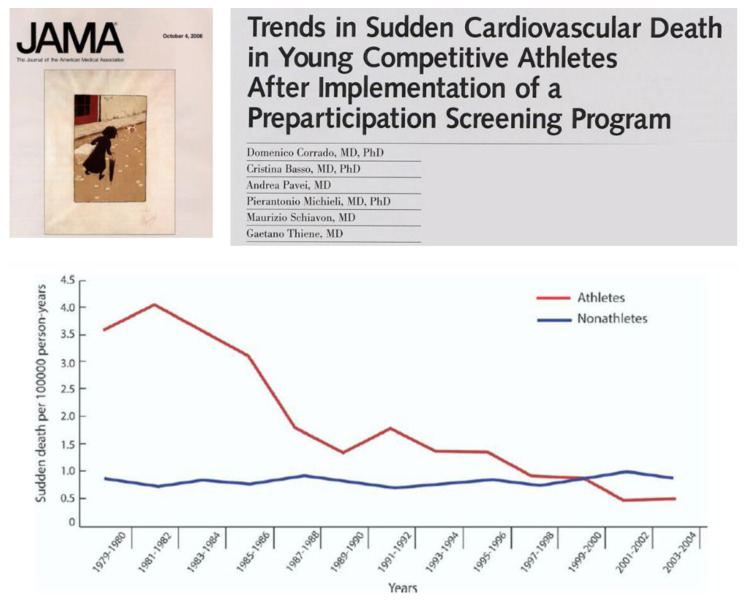
Sharp decline of sudden death in the young after introduction of ECG for screening for sports eligibility. From [38], modified.

**Figure 20 biomedicines-11-02018-f020:**
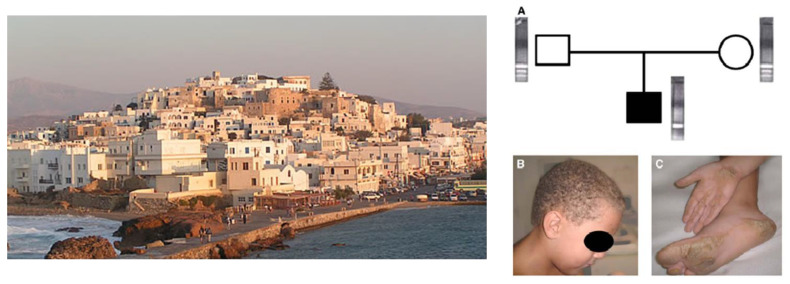
Recessive form of arrhythmogenic cardiomyopathy and woolly hair (cardiocutaneus syndrome) in the island of Naxos. Recessive transmission (**A**). Wolly hair (**B**). Palmo-plantar keratosis (**C**). From [23], modified.

**Figure 21 biomedicines-11-02018-f021:**
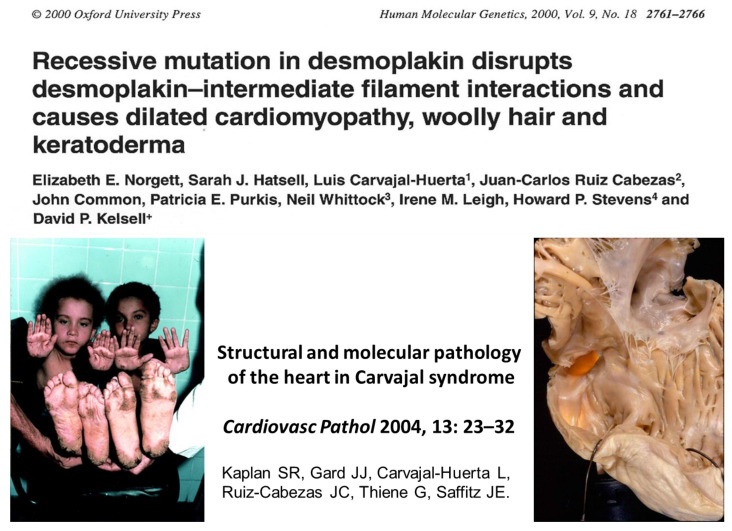
Carvajal syndrome, with recessive biventricular AC, due to a desmoplakin mutation [47]. Note a right ventricular aneurysm in the heart specimen.

**Figure 22 biomedicines-11-02018-f022:**
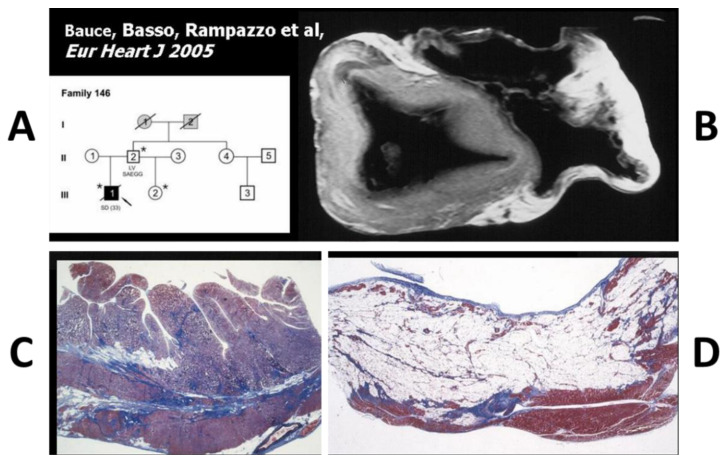
Desmoplakin mutation in dominant AC (**A**) with biventricular involvement (**B**–**D**). Azan Mallory stain, original magnification ×5. * Mutation carrier. From [49].

**Figure 23 biomedicines-11-02018-f023:**
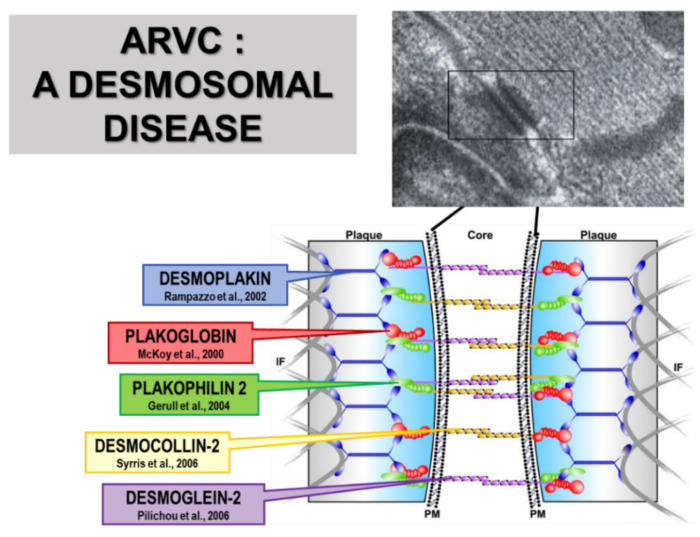
Mutations of proteins of the intercalated disk account for arrhythmogenic cardiomyopathy, giving rise to the name “desmosomal disease”. Transmission electron microscopy ×30,000. The rectangular includes the normal desmosomes, surrounded by disrupted ones. DESMOPLAKIN [48], PLAKOGLOBIN [44], PLAKOPHILIN 2 [50], DESMOCOLLON-2 [52], DESMOGLEIN-2 [51]. From [6].

**Figure 24 biomedicines-11-02018-f024:**
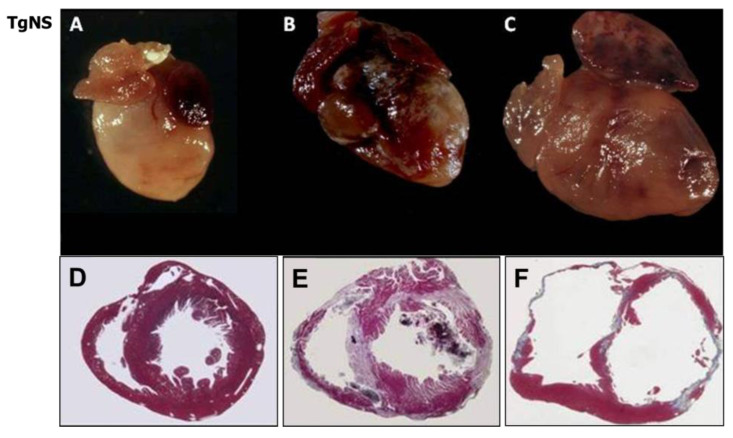
Gross (**A**–**C**) and histological (**D**–**F**) views of desmoglein transgenic mice. Note the progression of the disease with time. Azan Mallory stain, original magnification ×3. From [56], modified.

**Figure 25 biomedicines-11-02018-f025:**
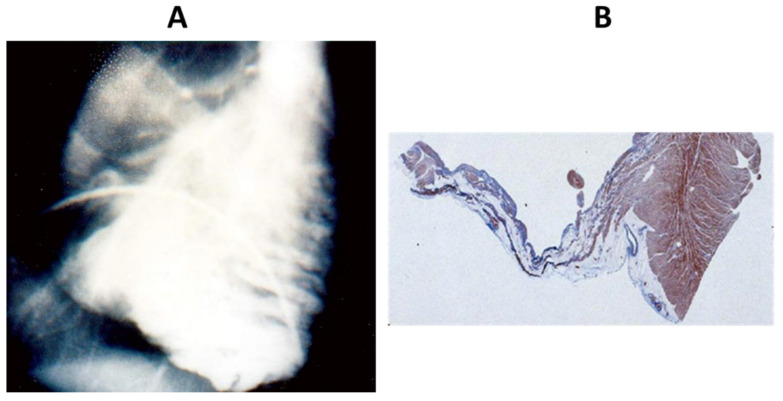
Triangle of dysplasia seen at angiocardiography (**A**) and compared with pathology (**B**). Azan Mallory stain, original magnification ×4. From [102], modified.

**Figure 26 biomedicines-11-02018-f026:**
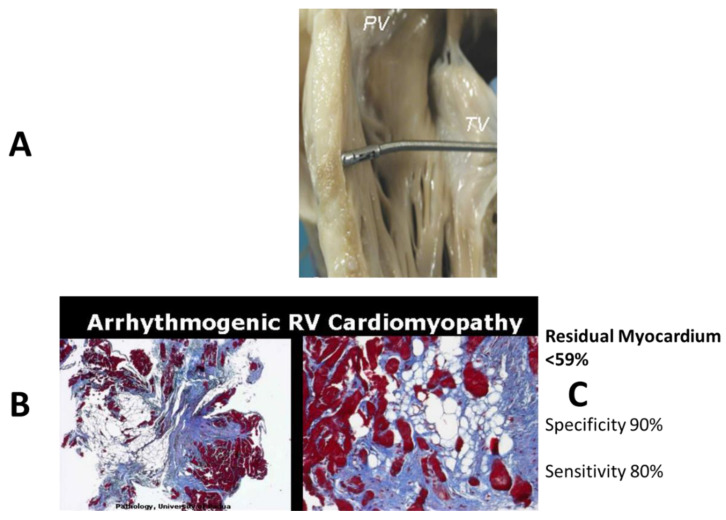
Right ventricular endomyocardial biopsy in AC. (**A**)The transmural fibro-fatty replacement favors the sensitivity of endocardial approach. (**B**,**C**) Histology revealed fibro-fatty specificity of the endomyocardial sampling. Azan Mallory stain, original magnification (**B**) ×20, (**C**) ×60. (**A**) From [103], modified; (**B**,**C**). From [23], modified.

**Figure 27 biomedicines-11-02018-f027:**
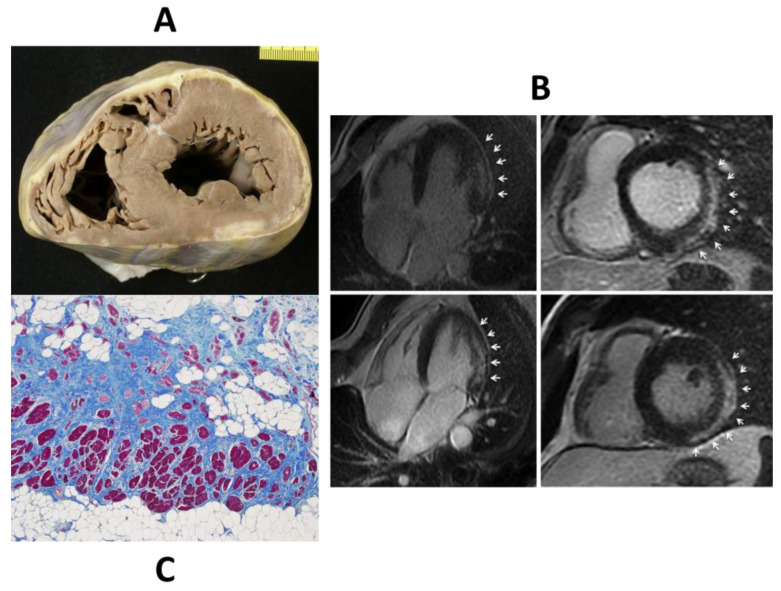
Left ventricular AC, seen either at gross (**A**) and histological study (**C**) and by cardiac magnetic resonance with gadolinium (**B**). Arrows indicate the fibro-fatty replacement of the left ventricle at late enhancement Cardiac Magnetic Resonance. Azan Mallory stain, original ×15.

**Figure 28 biomedicines-11-02018-f028:**
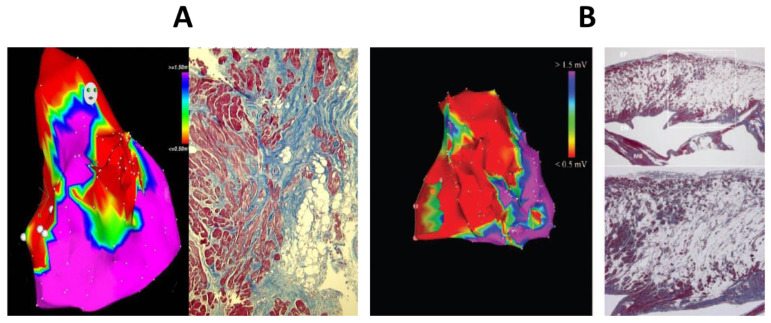
Electroanatomic mapping in AC (electrical scar) compared with endomyocardial biopsy (**A**) and autopsy finding (**B**). Azan Mallory stain, (**A**) ×15, (**B**) ×3 and ×10. (**B**) Right down is the amplification of the white rectangle box of the right up. (**A**) from [6]; (**B**) From [23].

**Figure 29 biomedicines-11-02018-f029:**
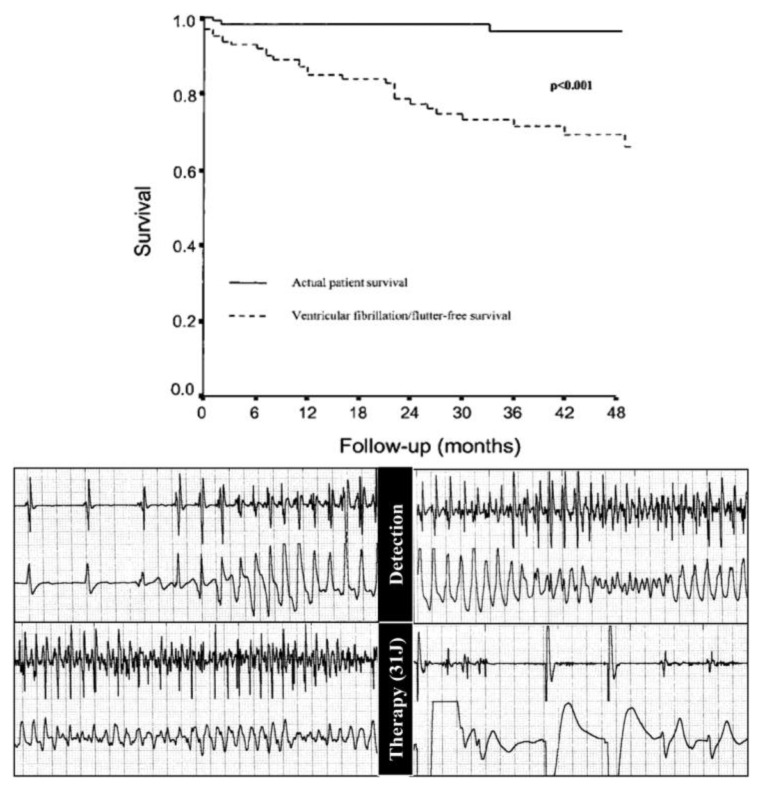
Efficacy of implantable cardioverter defibrillator (ICD) in AC. Survival vs. number of patients with electric shock. The difference accounts for number of saved lives. From [6].

**Figure 30 biomedicines-11-02018-f030:**
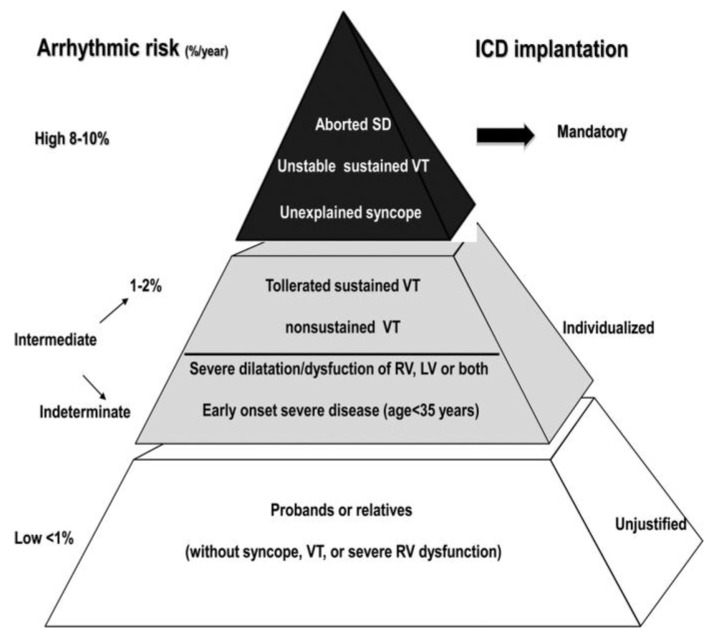
Risk stratification of arrhythmogenic cardiomyopathy with indication of ICD implantation. From [6].

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
