# Peer review of "Desmosomal Arrhythmogenic Cardiomyopathy: The Story Telling of a Genetically Determined Heart Muscle Disease"

_biomedicines, 2023, doi:10.3390/biomedicines11072018_

Round 1
Reviewer 1 Report
The manuscript 'Arrhythmogenic cardiomyopathy: the story telling of a genetically determined heart muscle disease' submitted by G. Thiene is interesting but needs several modifications and updates.
1.) The clinical description of ACM by Lancisi is realling interesting. However, maybe it should be underlined that this clinical description was published before Gregor Mendel performed his grand breaking genetic experiments.
2.) The fibro-fatty replacement is historically summarized. However, novel techniques like e.g. spatial transcriptomics were recently used to characterize the degeneration of cardiomyocytes in ACM. Could the author discuss this novel finding in the human explanted heart of a PKP2 mutation carrier.
3.) The paragraph ‘AC: a hereditary genetic disease’ needs several updates and extensions:
Today, there are more genet known that the classical desmosomal genes. What is with DES, PLN, LMNA, LEMD2, ILK, TMEM43… Could you shortly discuss these other non-desmosomal genes including relevant references?
In addition, the author should indicate examples for a recessive inheritance by homozygous or compound heterozygous mutations for example in DSC2 and DSG2. (A homozygous DSC2 deletion associated with arrhythmogenic cardiomyopathy is caused by uniparental isodisomy.) (Hemi-and homozygous loss-of-function mutations in dsg2 (Desmoglein-2) cause recessive arrhythmogenic cardiomyopathy with an early onset.")
What is with de novo mutations? For example it was shown, that a de novo mutation in the DES gene cause ACM. (DES-N116S). Could you discuss de novo mutations and the relevance of genetic analysis even if there are no further family members present with ACM.
What is with founder mutations. Could you discuss this?
Could you please include a short paragraph about different animal models used for ACM. In minimum, a recent review article summarizing animal models for ACM should be referenced.
4.) Could you include a paragraph about modern gene therapy methods like CRISPR-Cas9 or AAV gene therapy which maybe useful for therapy development in the future?
5.) Several figures are presented without scale bars (e.g. Figure 12/13/14 and several more). Could you insert scale bars.
6.) Are all figures really relevant for this manuscript? I have the feeling that several figures are personal private figures. The author should verify if they are really necessary to understand the objective story of ACM.
In summary, I think the submitted manuscript has strengths in the clinical description. The genetic part needs several updates and needs expansion and extensions since novel literature is mainly ignored. In total, I suggest a major revision for this manuscript.
The English should be double checked by a native speaking editor.
Author Response
We believe that there was a misunderstanding. We have been asked to write a chapter for a Special Issue "Advanced Research in Arrhythmogenic Cardiomyopathy" of Biomedicines (section "Molecular and Translational Medicine"), reporting the history of discoveries and applied inventions on AC in modern and contemporary history.
We focused our attention to desmosomal AC, where we have focused our research experience.
Accordingly, the title of the paper has been changed (Desmosomal arrhythmogenic cardiomyopathy: the story telling of a genetically determined heart muscle disease).
Answers to Comments and Suggestions for authors
1.) Gregor Mendel contribution was added after the first clinico-pathological description by Lancisi.
2.,3.,4.) Very recent updates are out of the scope of our review, mainly devoted to classical desmosomal AC. They will be part of other chapters of this special issue, devoted to overlapping syndromes, novel genetics and different animal models. We don’t want to invade the field of other authors.
5.) According to reviewer’s suggestions we added amplification number to all histological and transmission electron microscopy figures.
6.) We removed personal photos of the authors.
Comments on the Quality of English Language
The English will undergo to an editorial improvement: it will take time.
Reviewer 2 Report
We read with interest the narrative review by Dr. Thiene describing the history, genesis, diagnosis, and genetic basis of Arrhythmogenic cardiomyopathy.
the narrative is highly informative because it involves discovery and case studies in Arrhythmogenic cardiomyopathy.
Comments:
The only comments are related to the writing style and the need to include a helpful "timeline of the dates relating Arrhythmogenic cardiomyopathy to discovery and its development" that would be so helpful for the readers.
Many of the mentioned figures can be collated together as one figure, the use of a timeline can reduce the historical information. Also, a final note is that there should be a section for future direction and challenges and also the advanced technology application such as Omics.
Minor Comments:
English writing needs extensive editing as many sections are written in a fragmented way and needs restructuring in terms of grammar and paragraph style.
English writing needs extensive editing as many sections are written in a fragmented way and needs restructuring in terms of grammar and paragraph style. Many paragraphs are only made of one sentence.
Author Response
We thank the referee for the congratulations to our narrative, informative review.
Comments:
- We added a table reporting the historical “timeline of the dates relating AC to discovery and its development”. We thank the reviewer for the suggestion: it will be an added value to our contribution.
- We removed personal figures and collected some others.
Minor Comments:
The English language will undergo to editorial improvement.
Round 2
Reviewer 1 Report
Dear editors and authors,
I have really a severe problem with this manuscript. As I have written in my last review this manuscript needs more objective story telling about ACM. The authors concentrate in the majority on their own research. This cause really a high rate of self-citates (more than 30 self citations). This are nearrly half of all citations. In addition, this concentration on their own research cause a subjective picture of ACM. The genetic paragraph is really summarizing only a small part of the knowdlege about the genetic background of ACM. Many novel papers published by other authors were completely ignored. In this personal style, I think the review article is misleading many readers. Therefore I stronlgy suggest a rejection of this article. I hope four your understanding!
The English needs several changes and corrections.
Author Response
Comments and Suggestions for Authors
Dear editors and authors,
I have really a severe problem with this manuscript. As I have written in my last review this manuscript needs more objective story telling about ACM. The authors concentrate in the majority on their own research. This cause really a high rate of self-citates (more than 30 self citations). This are nearrly half of all citations. In addition, this concentration on their own research cause a subjective picture of ACM. The genetic paragraph is really summarizing only a small part of the knowdlege about the genetic background of ACM. Many novel papers published by other authors were completely ignored. In this personal style, I think the review article is misleading many readers. Therefore I stronlgy suggest a rejection of this article. I hope four your understanding!
We regret, we believed that our commitment was to write a historical review article on desmosomal arrhythmogenic cardiomyopathy (AC).
We fully agree on the need to report also non desmosomal AC, including genetics. Otherwise the review may appear self-referential. To this purpose, we added a chapter to address this weakness, with appropriate references.
Comments on the Quality of English Language
The English needs several changes and corrections
A thorough, extensive revision of the English was performed by a native British.
Round 3
Reviewer 1 Report
The authors have significantly improved their manuscript. I have only one small point:
In the chapter about non-desmosomal genes causing ACM, the authors should include ILK (encoding integrin linked kinase, p.H33N and p.H77Y) and LEMD2 (LEM domain nuclear envelope protein 2, p.L13R) aus ACM causing genes including relevant references.
The manuscript should still be corrected by a native English speaking editor.
Author Response
Comments and Suggestions for Authors
The authors have significantly improved their manuscript. I have only one small point:
In the chapter about non-desmosomal genes causing ACM, the authors should include ILK (encoding integrin linked kinase, p.H33N and p.H77Y) and LEMD2 (LEM domain nuclear envelope protein 2, p.L13R) aus ACM causing genes including relevant references.
We thank the referee to acknowledge the significant improvement of the manuscript.
We thank for the suggestion: in the chapter Non-desmosomal Arrhythmogenic Cardiomyopathies we added ILK and LEMD2 as ACM causing genes, with relevant references.
Comments on the Quality of English Language
The manuscript should still be corrected by a native English speaking editor.
The English has been further checked by a colleague fluent in English writing.